# Analysis of Usage Data from a Self-Guided App-Based Virtual Reality Cognitive Behavior Therapy for Acrophobia: A Randomized Controlled Trial

**DOI:** 10.3390/jcm9061614

**Published:** 2020-05-26

**Authors:** Tara Donker, Chris van Klaveren, Ilja Cornelisz, Robin N. Kok, Jean-Louis van Gelder

**Affiliations:** 1Department of Clinical, Neuro and Developmental Psychology, Section Clinical Psychology, Vrije Universiteit Amsterdam, Van der Boechorststraat 1, 1081 BT Amsterdam, The Netherlands; 2Amsterdam Public Health Research Institute Amsterdam, Van der Boechorststraat 7, 1081 BT Amsterdam, The Netherlands; 3Department of Psychology, Laboratory of Biological and Personality Psychology, Albert Ludwigs-University of Freiburg, Peter-Kaplan Meierstrasse 8, 79104 Freiburg im Breisgau, Germany; 4Department of Education Sciences, Section Methods and Statistics and Amsterdam Center for Learning Analytics, Vrije Universiteit Amsterdam, Van der Boechorststraat 1, 1081 BT Amsterdam, The Netherlands; c.p.b.j.van.klaveren@vu.nl (C.v.K.); i.cornelisz@vu.nl (I.C.); 5Department of Psychology, University of Southern Denmark, Campusvej 55, 5230 Odense, Denmark; rkok@health.sdu.dk; 6Centre for Innovative Medical Technology, Odense University Hospital, Indgang 101, 5000 Odense, Denmark; 7Institute of Education and Child Studies, Leiden University, Pieter de la Court building, 4th floor, Wassenaarseweg 52, 2333 AK Leiden, The Netherlands; jlvangelder@gmail.com; 8Max Planck Institute for the Study of Crime, Security and Law, Department of Criminology, Günterstalstraße 73, 79100 Freiburg, Germany

**Keywords:** acrophobia, cognitive behaviour therapy, mobile app, virtual reality, usage data

## Abstract

This study examined user engagement with ZeroPhobia, a self-guided app-based virtual reality (VR) Cognitive Behavior Therapy for acrophobia symptoms using cardboard VR viewers. Dutch acrophobic adults (*n* = 96) completed assessments at baseline and immediately following treatment. Primary outcome measures were the Acrophobia Questionnaire (AQ) and the Igroup Presence Questionnaire (IPQ). Usage data consisted of number of VR sessions practiced, practice time, and fear ratings directly after practicing. Results show that of the 66 participants who played at least one level, the majority continued to finish all levels, spending on average 24.4 min in VR. Self-reported fear consistently decreased between the start and finish of levels. Post-test AQ scores depended quadratically on time spent in VR. Higher pre-test AQ scores were significantly associated with subjective anxiety after the first level and a reduction of post-test AQ scores, but not with number of sessions, suggesting it might be more beneficial to play one level for a longer time period instead of practicing many VR levels. Results also show an optimum exposure level at which increasing practice time does not result in increased benefit. Self-guided VR acrophobia treatment is effective and leads to consistent reductions in self-reported anxiety both between levels and after treatment. Most participants progressed effectively to the highest self-exposure level, despite the absence of a therapist.

## 1. Introduction

Given the global challenge of access to evidence-based psychological treatment for common mental health disorders, there is an evident need for affordable and scalable self-help interventions [1]. Reasons for limited access include a lack of mental health professionals and high treatment costs [2]. Digital interventions may offer a solution. Several meta-analyses of randomized controlled trials (RCTs) have demonstrated effectiveness for digital interventions, mostly online, for treating common mental disorders such as anxiety disorders [3,4,5]. Innovations in this field comprise virtual reality (VR) and mobile applications (apps) [2]. There is empirical evidence that such interventions can be similar in effectiveness compared to face-to-face treatment [1,6,7,8,9,10]. For example, Morina et al. [9] demonstrated in their meta-analysis that results of behavioral assessment at post-treatment and at follow-up revealed no significant differences between VRET and exposure in vivo. Furthermore, in our study [1] we showed that results of an VR-CBT-based app were comparable to results found in studies investigating the effectiveness of traditional CBT. However, less is known about how user adherence and engagement relates to effectiveness of these interventions, especially with regard to VR and mobile app-based interventions. One exception is the study of Hong et al. [11] in which the effectiveness and usage of a mobile-based self-training VR program for acrophobia was investigated. Interestingly, heart rate (HR) and gaze down percentage were also included in this study. Using a pre-post study design with two arms (high and low acrophobic symptoms), Hong et al. demonstrated that participants with higher acrophobia symptoms derived more benefit from the VR program compared to those with lower acrophobia symptoms. Furthermore, they found a negative correlation with gaze-down percentage in the high acrophobia symptom group compared to the low acrophobia symptom group. In this study, subjects attended a VR-center and had contact with research staff. Treatment efficiency and effectiveness could be enhanced by a better understanding of how intervention usage affects outcome [12,13,14]. For example, treatment efficiency can be improved through the identification of redundant elements that do not contribute to symptom reduction. Treatment effectiveness can be increased by identifying where participants drop-out and improving that element to reduce drop-out. Exploring usage data of fully self-guided interventions is of particular importance due to the lack of human oversight and the resulting inability to adjust the course of such interventions [12]. An analysis of usage data can be beneficial in optimizing the uptake and continued use of these self-guided interventions.

Recent studies examining usage data of digital interventions have demonstrated that highly active users completing more modules predict better outcomes for eating disorders, smoking cessation and depression [15,16,17,18,19]. Previous studies also demonstrated that higher activity during the first week of treatment is a predictor of better adherence for web-based interventions [20,21]. Moreover, more concise and shorter interventions achieve better usage rates compared to extensive interventions [22]. However, in a recent review, Donkin et al. [15] concluded that several potential usage metrics (number of log-ins, time spent online) were inconsistently associated with outcomes for online interventions for psychological disorders. Only the relation between proportion of completed modules and outcome emerged as a consistent association [15,16]. Previous research into VR treatment indicated that presence plays an important role in the effectiveness of a program [23]). For example, in our study reporting the main results of this trial we found that a larger reduction in acrophobia symptoms was associated when the feeling of being present in the virtual environment was higher [1].

The efficacy of VR interventions for acrophobia has been well-documented (for an overview, see [24]. However, with few exceptions (e.g., [25]) usage data of virtual reality (mobile app) interventions for common mental disorders has remained largely unexplored. For self-guided VR interventions, participant retention is important as there is no therapist oversight into the process. The aim of the present study was to examine usage of, and engagement, with, ZeroPhobia, a fully self-guided app-based virtual reality Cognitive Behavior Therapy (VR-CBT) for acrophobia, using mobile phones and a low-cost (cardboard) virtual reality viewer, and to determine user metrics contributing to effectiveness. The most important element of CBT for anxiety disorders is exposure, in which the participant is repeatedly confronted with the feared object or situation, thus learning that the expected disaster is not happening. This, in turn, leads to decreases in anxiety symptoms [26]. We specified the following exploratory hypotheses: (1) higher VR activity (number of completed VR sessions, practice time in VR) is associated with reduced post-test acrophobia symptoms; (2) higher presence scores on the IPQ are correlated with stronger decreases in anxiety ratings directly after practicing with exposure in the VR environment; (3) higher acrophobia symptoms at pre-test correlate positively with VR anxiety ratings and VR activity, and negatively with a reduction in acrophobia symptoms at post-test; and (4) post-session anxiety levels consistently decrease compared to pre-session anxiety levels after repeated practice in the VR environment.

## 2. Materials and Methods

### 2.1. Study Design and Procedure

In the current study, we carried out a secondary analysis of a previously published outcome study [1]. Details of the materials and methods are described elsewhere and will therefore not be dealt with in detail here [1,27]. In short, in this single-blind RCT, participants were recruited from the Dutch general population through websites, magazines and local media. Ethical approval was received from the Medical Ethics Committee of the VU University Medical Center (registration number 2016-563, Trial registration: NTR6442) [1]. Participants were randomized into two groups: intervention or waitlist. The research team was blind to treatment allocation. All materials were completed online without researcher intervention. Trial Registration: Nederlands Trial Register http://www.trialregister.nl identifier: NTR6442 (prospectively registered).

### 2.2. Participants

Participants (18–65 years) who provided written informed consent by email or mail, who scored at least 45.45 on the Acrophobia Questionnaire (AQ)-Anxiety [28,29], had access to an Android smartphone (Android v.5.1 Lollipop or higher, 4.7–5.5 inch screen and gyroscope) were included in the study. Participants with insufficient Dutch language skills, or participants receiving current phobia treatment or psychotropic medication < 3 months were excluded from the study, as well as participants having severe depression (Patient Health Questionnaire [PHQ-9], [30]; total score > 19) or suicidality (Web Screening Questionnaire; WSQ, score ≥ 3; [31]). Enrollment commenced 24 March and ceased 28 September 2017.

### 2.3. Intervention: ZeroPhobia

ZeroPhobia-Acrophobia consists of six animated and engaging modules using 2D animations which provide background information and explain key concepts (e.g., the fear curve). The annimations are accompanied by an explanatory voice-over and an animated virtual therapist (modelled after the first author) about the nature and origins of the phobia, how to deal with it, setting goals, exercises, getting through difficult moments, cognitive behavioral therapy to deal with negative thoughts and practicing with challenging situations. The modules take between 5 and 40 min to complete. Exposure, the core of the treatment, is realized through gamified mobile VR and four 360° videos covering the entire acrophobia exposure spectrum. Participants started using the VR and the 360° videos from Module 3 onwards and navigated through the virtual environment using gaze control. Gaze control is a method for the hands-free selection of objects and the activation of functions within a virtual environment. By looking at an object or button in the center of the field of view for a specified period of time, e.g., 2 s, the object or button is selected or activated. In ZeroPhobia, large arrows served as interactive buttons used for navigating the virtual environment. By gazing at an arrow, a user moved to the location of the arrow. Similarly, items that needed to be collected, as part of the assignments in the various levels, could be selected by looking at them. The cardboard viewer could be strapped to the head. The VR involved a gamified virtual theater. In the game participants had to complete a series of increasingly challenging tasks (e.g., changing a light bulb on a small ladder, connecting speakers at the edge of the stage, going up a high ladder to repair a small damaged platform, fixing a spotlight on the highest balcony, saving a cat while being on a gangway high above the stage). In each level, participants had to look at assets located on the theatre floor that needed to be collected, hence encouraging them to look down and face their fears [27]. For more details on the VR environments used, see [1,27].

Because any VR setup that generates frame rates below 90 frames per seconds is likely to induce disorientation, nausea and other negative user effects, the frame rate was increased to an optimal level to minimize the risk of cybersickness. Furthermore, there were no quickly moving objects and it was not possible for the user to move quickly through the VR environment. Battery drainage was reduced by keeping the VR levels at an optimum duration meaning that playing a VR level did not exceed 10 min, although participants were at liberty to practice in VR as long as they felt like they needed. Battery drainage was further for small degree lessened indirectly by removing the back cover off the phone, which reduced overheating. The app provides safety instructions to participants before entering the VR environment. For example, participants were instructed to remove all sharp objects in their environment prior to commencing with VR-exposure, to avoid possible injuries. Also, they were instructed to start practicing in VR while seated. Only once anxiety decreased, participants were encouraged to practice standing up. They were also instructed to take off their VR viewer immediately if they felt they might fall.

For the current usage analyses, only information from the interactive VR environment was used, not the 360° videos. This is because the majority of participants were unable to view the 360° videos due to technical limitations of their smartphone (they encountered a black screen). They could, however, view the 360° videos on YouTube. The trial was delivered over a 3-week period during which participants were at liberty to practice with ZeroPhobia as often and as long as they wanted [1]. Weekly standardized motivational e-mails with reminders to start or continue with ZeroPhobia were sent to participants during the intervention period. For details, see [27] and for ZeroPhobia screenshots Appendix A. The VR environment was created with the Unity game engine (version 2017.3.0f3; Unity Technologies, San Francisco, CA, USA).

### 2.4. Outcomes

All questionnaires were completed online. Measures were taken at baseline (pre-test), immediately after the intervention (posttest), and 3 months after the intervention (follow-up). Participant characteristics measures were collected at baseline, while symptom measures were administered at each time point. All assessments were programmed with Survalyzer software [32]. See [27] for details on outcome measures. The primary outcome was the 20-item Acrophobia Questionnaire (AQ) [28]. The AQ is a widely-used validated instrument [29]. The anxiety subscale is measured using a 7-point Likert scale (0 = not anxious to 6 = extremely anxious). Total score ranges is 0–120. The avoidance subscale uses a 3-point Likert scale (“I would not avoid it” to “I would not do it under any circumstances”). Secondary outcomes included in this study were the Igroup Presence Questionnaire (IPQ); [33] to assess presence in VR, and usage. Because the IPQ is widely used in VR research and to be able to compare results on presence with previous studies, we chose the IPQ instead of other measures targeting presence. Usage data consisted of practice time in VR (for each level [time between entering and exiting the VR level] and in total [the sum of all practice time in VR per patient in minutes]), number of sessions (were one session is defined as repeatedly practicing with the same level) and anxiety ratings directly after practicing a session in the VR environment. As described in [27], participants were encouraged to progress to a more difficult level as soon as self-reported fear dropped below 4 on a 10-point scale. For self-reported fear between 4–7, participants were advised to try the same level again, and for self-reported fear levels 8–10 they were strongly advised to keep practicing the current level. Participants could not continue to the next level without self-reporting fear under 4.

### 2.5. Usage Data Retrieval

Usage data was retrieved from the ZeroPhobia app to a server which then stored the data on a database, on a secure (SSL) website. All communication between the app and the database was encrypted by means of a certificate. To prevent others (non-participants) from contaminating the database with data, adding new data was only possible by sending the correct key from one of the participants in the study. The data was pseudo-anonymized, meaning that data was anonymized but linked with the trial identifier consisting of four numbers. Examples of database reads are time spent in a VR level, time stamps (start and end time of a participant in the VR environment) that enabled us to exactly determine duration of practice time for each VR level, and experienced anxiety levels after each VR level played.

### 2.6. Statistical Analyses

Demographic and clinical characteristics are presented in terms of means and SDs. Usage data are presented in terms of means, SDs, and minimum and maximum observations for two groups: (1) all participants and (2) participants who have at least experienced one VR session. The primary outcome (AQ) was replicated using the same imputation methods and covariates as performed in Donker et al. [1], with the exception that the model was not estimated with OLS but instead with Maximum Likelihood Estimation. This estimation method is convenient as it allows parameterization of the treatment effect such that associations with the usage covariates can be directly estimated. Two-sided *p*-value < 0.05 indicated statistical significance. STATA version 14.2 (StataCorp LP., Texas, TX, USA) were used for the analyses. A data monitoring committee was not required by the Ethics Committee because of the expected low safety risk of the participants.

## 3. Results

### 3.1. Sample, Baseline Characteristics and Cybersickness

Details of the participant flow and drop-out are described elsewhere [1,27]. In short, of 663 individuals who signed up for participation, 291 were ineligible (e.g., due to phone ineligibility) and therefore excluded from participation. In total, 193 participants filled in the baseline assessment and were randomly assigned to the VR-CBT app condition (*n* = 96) or to the wait-list control condition. The pre-treatment attrition rate was 23% in the app condition because of illness (1 [1%]) or an incompatible smartphone (21 [22%]) [1]. Of the total sample (N = 96), the mean age was 41 years (SD = 13.73) and 66 (68.75%) of them were female. Most participants completed postsecondary education (*n* = 84; 87.5%). The mean AQ baseline score was 85.16 (SD: 18.42). Of the 96 randomized participants, 21 (23%) could not download ZeroPhobia on their smartphone because the smartphones were lacking a gyroscope (required for experiencing VR) and one (1%) did not start ZeroPhobia because of illness, leaving the sample with *n* = 74 participants.

### 3.2. General VR Usage Data

Of the 74 participants, 66 participants (68.8%) experienced at least one VR session. Usage data for these 66 participants are shown in Table 1, where the total VR duration time of active users is 24.4 min, with a minimum of 0.6 min (36 s) and a maximum of 71.05 min. On average, active participants practiced with around nine VR sessions. As can be seen, the anxiety ratings of participants are substantially different for practicing with level 1 compared to the highest level they attained (for most participants, this was level 5). The anxiety rating after playing a more difficult level, was on average lower than the anxiety rating after playing level one for the first time. When the anxiety ratings after playing a certain level for the first time were compared with anxiety ratings after playing a certain level for the last time (e.g., the anxiety rating after having played level 4 for the last time as compared to playing level 4 for the first time), we noted that anxiety was reduced by around 1.3 points on average (range: 1–10).

A paired *t*-test indicated that this difference was statistically significant (*t* (280) = 12.28, *p* < 0.0001) (see Table 2).

### 3.3. Level-Specific Usage Data

As can be seen in the level-specific usage results in Table 2, the self-reported anxiety of users was, on average, relatively low (M = 4.302, SD = 2.397) when finishing levels for the first time (initial anxiety). The final anxiety numbers show that on average, users exited a level with a self-reported anxiety level of 2.665 (SD = 1.270). This shows that most users complied well with the advice to ’level up’ after self-reported anxiety had dropped below 3, however some users chose to replay a level up to 11 times (level 4).

### 3.4. Replication of Main Results

An intent-to-treat analysis showed a significant reduction of acrophobia symptoms at post-test at 3 months for ZeroPhobia compared with the controls (*b* = −26.73 [95%CI, −32.12 to −21.34]; *p* < 0.001; *d* = 1.14 [95%CI, 0.84 to 1.44]). Using the ML-estimation procedure, similar results for ZeroPhobia effectiveness were found compared to original study (b(SE) = −26.7 (2.73), noting a slight difference in standard error due to using a ML-estimation rather than an ordinary least squares approach as in [1].

### 3.5. Hypothesis 1: More VR Activity Is Associated With Lower Post-Test AQ Scores

To test the hypothesis that more VR activity is associated with a greater reduction in AQ scores at post-test, we modelled the post-test scores of the AQ with the pre-test scores on the AQ, the number of sessions played and a linear and quadratic parameter of practice time in minutes. The results show that practice time and the number of sessions result in lower post-test AQ scores, although number of sessions is not statistically significant. Post-test AQ scores depends quadratically on time spent practicing (Practice Time: b(SE) = −1.07(0.391); *p* < 0.05; *PT*^2^: b(SE) = 0.018(0.005); *p* < 0.05). The association with number of sessions was not statistically significant. To test for robustness, this analysis was performed with and without baseline covariates in the model, but results did not change significantly. With respect to the estimated association between the number of sessions and Post-test AQ scores, we note that the inclusion of background characteristics does not change the estimated coefficient, but it does make the association insignificant. The insignificant result may be the result of lack of statistical power.

In Figure 1 the association between decrease in AQ scores and practice time and number of sessions is represented graphically, where darker areas denote a greater reduction in AQ scores at post-test. These results suggest the existence of a ’sweet spot’, an optimum level of exposure at which increasing practice time does not result in an increased benefit and that it does not significantly depend on the number of sessions. The optimum level of practice time in the VR environment was found to be 25.5 min.

### 3.6. Hypothesis 2: A Higher Level of Presence Is Associated with Greater Decrease in Post-Session Anxiety

To test this hypothesis, we tested the association between the IPQ scores post-test and the treatment effect while controlling for baseline covariates. We also controlled for practice time, and pre-test AQ scores, as this would be associated with treatment effect. Because not all participants filled in the IPQ at post-test (12/66 missing, 21.2%), this variable was also modeled.

The results showed that those who filled out the IPQ, a higher level of presence was associated with a larger reduction of AQ scores post-test b(SE) = −0.914 (0.225), *p* < 0.001, confirming our hypothesis. The addition of the IPQ variable to the model did not change the parameter for the AQ pre-test scores, indicating that presence is not associated with pre-test AQ scores. The practice time coefficient did change after the introduction of presence into the model, indicating that presence and practice time covary, which is evidenced by the high correlation between presence and linear and quadratic play time (*r* = 0.599 and *r* = 0.481 respectively).

### 3.7. Hypothesis 3: Associations between Scores

We hypothesized that the AQ pre-test scores were significantly associated with VR anxiety ratings after the first session. Moreover, we hypothesized that the AQ pre-test scores would be significantly associated with a reduction in acrophobia symptoms, but not with VR exercise time. The results showed that the pre-test AQ scores were significantly associated with the subjective fear rating after the first session (r = 0.134, *p* <0.05) and a reduction in AQ scores at post-test (r = −0.312, *p* < 0.05). However, contrary to our hypothesis, AQ pre-test scores were not associated with either linear practice time (r = 0.006, *p* >0.05) or quadratic practice time (r = −0.003, *p* > 0.05).

### 3.8. Hypothesis 4: Repeated Activity in a VR Level Leads to a Consistent Decrease in Post-Level Anxiety

We hypothesized that, consistent with general expectations about exposure therapy, self-reported fear would show consistently lower scores after a session when compared to before a level, since repeated exposure should result in lower fear levels after a session. As can be seen in Table 3, this is the case for each level, with an average drop of 1.35 points (SD 2.24) per level. As seen in the minimum and maximum scores, some participants exit levels with higher fear ratings than they started with, indicating unsuccessful exposure exercises.

## 4. Discussion

To our knowledge, this is the first study to look at self-guided VR exposure on a session-to-session basis. As self-guided exposure relies strongly on the participant to guide the progress through increasingly more challenging sessions, the inter-session data provides relevant information to guide future development of similar interventions. Moreover, successful exposure therapy relies on adequately modulating how challenging the different sessions are. This makes it important to verify that successful exposure took place within a session—as witnessed by a decreased anxiety level after a session as compared to before a session—in the absence of a therapist to guide this. Consistent with expectations from exposure therapy, the results indicate that overall, participants reported a decrease in fear as they progressed through the levels. It should be noted, however, that for some participants the exit scores remained relatively high, indicating unsuccessful exposure. The results further demonstrated that post-test AQ scores depended quadratically on time spent practicing, but not with the number of sessions practiced, indicating that it might be more important to play one level for a longer period of time instead of practicing many VR levels. Moreover, it was shown that, in line with previous research, a higher presence was associated with better outcomes [23].

Most participants advanced through all five of the VR exposure levels, indicating that—even in the absence of a therapist—self-guided VR therapy in the home setting is feasible and can be effective, even with rudimentary equipment such as a cardboard VR viewer and the participants’ own smartphones. Importantly, the relatively high number of participants who completed all levels shows that the app was engaging and convincing enough to persuade participants to keep practicing. This motivational aspect is reflected by the number of sessions participants practiced, where on average participants tried each level at least twice, and in some cases up to 11 times. One explanation is that the threshold set for going to a subsequent level through levels (a self-reported fear of less than 4) added an element of challenge to the exposure, motivating participants to keep trying a level to unlock the next [34]. However, evidence for the success of gamification in smartphone apps is currently lacking [35].

Although existing VR interventions have been envisaged as solutions to be used in guided form under the supervision of a therapist (see e.g., [24]), ZeroPhobia was designed as a fully self-guided intervention with no therapist oversight. Therefore, it was crucial to monitor participant progression to verify whether successful in-virtuo exposure was taking place. When practicing with exposure exercises, it is important to correctly pace the increase in difficulty and challenge. Progressing too quickly through levels could result in excessive anxiety, leading to dropout, while practicing at an easy level for too long is inefficient and may lead the participant to disengage because it is not challenging enough. In ZeroPhobia it seems that these decisions were in line with the recommendations on pacing progress through the levels as shown in the app. In Table 2 it is shown that the average final anxiety scores in the levels were all below 4, since participants could not progress unless their self-reported fear had dropped under 4. This is visible in the self-reported fear levels after practicing a level for the first time. These averages also show that participants tended to ’level up’, and not unhelpfully linger in levels. As most participants progressed to the highest level of the VR exposure game (level 5), it was not possible to find a reliable cut-off value for number of levels that should be completed, or to find a level that should minimally be attained for deriving clinical benefit; but the corollary is that participants chose to persist with the exposure exercises until the end, indicating successful and perhaps even enjoyable exposure experiences. However, we did find that increased activity was related to lower AQ scores at post-test. Results also demonstrated an association between decrease in AQ scores and practice time and number of sessions, suggesting the existence of a ’sweet spot’, an optimum level of exposure at which increasing practice time does not result in increased benefit. In this study we found that participants derive most benefit when the practice time in the VR environment is 25.5 min irrespective of the amount of VR sessions. However, these results need to be interpreted with caution because it is unknown how many participants practiced for how long with the 360 videos.

The results showed that the pre-test AQ scores were significantly associated with the subjective fear rating after the first session and a reduction in AQ scores at post-test. This would suggest that those with higher acrophobia symptoms could derive more benefit from the VR exposure. This is also in line with previous research findings [11]. It could however also be a floor effect, where participants with already relatively low acrophobia symptoms have relatively little to gain from the intervention. Regardless, the results show that the VR environment offers a safe and engaging way to self-guided exposure, even for those with more severe complaints.

### Limitations

The current study had several limitations. Firstly, all the hypotheses in this study were exploratory in nature, leading to the possibility of false positive findings. Secondly, more granular data in terms of logins and time spent in app modules would have been helpful for a more detailed view of user activity within the VR sessions. Thirdly, we did not have access to usage data of the 360 videos from YouTube. This means that several participants have practiced with 360° video exposure which could also contribute to a decrease in anxiety ratings at post-test. The optimum level of exposure time is a conservative estimate therefore. Fourthly, we did not systematically conduct data on the performance of the cardboard VR viewer during the sessions, therefore we are unable to evaluate the interaction of the performance of the VR viewer on its usage and effectiveness. However, only one participant contacted the research team to request a new viewer because the one originally provided to them had broken. Fifthly, due to a small sample size, some insignificant result may be the result of lack of statistical power. The great inter-subject variability in usage is remarkable. It might be that, depending on the personal history, VR stimulation will work at one moment of exposure, without a tool to identify the triggering event, conditioning success of fear reduction. Sixthly, data quality can be a problem in an uncontrolled environment, when only relying on self-report measures and no biological measures. However, research has demonstrated that self-report measures can have good to excellent validity when compared to a diagnostic interview [36]. Furthermore, participants can be more honest with filling in the questionnaires, as the computer has no ’eyebrows’ [37]. Moreover, in a meta-analysis targeting VR interventions, Morina et al. [9] concluded that the behavioural measurement effect sizes were similar to those calculated from self-report measures in the VR studies, indicating no differences between the two types of measurements. Furthermore, we have conducted robustness analysis which confirmed that the VR-CBT ZeroPhobia app had a strong impact on the anxiety for heights, even when analyzed very conservatively in a randomized controlled design, and that the general anxiety effect did not drive the results. Lastly, as yet we have no data on external validity, especially on whether the effects of the VR exposure translate to decreased acrophobia and acrophobic avoidance in real-world settings, and commensurate increases in quality of life.

Future research is needed to replicate the main results of the study, especially with regard to external validity and translational effects in real life. Furthermore, more granular data on, e.g., eye gaze could generate valuable information on which cues participants choose to engage with, as even in VR participants can choose to look away from the fear-inducing stimulus. It would also be interesting to investigate whether adding sound to the VR experience effects the feeling of presence.

## 5. Conclusions

In sum, our findings show that self-guided VR for acrophobia symptoms is feasible, effective, and follows the same general patterns in terms of self-reported fear reductions as one would expect from therapist-guided exposure exercise. The results indicate that participants were engaged with ZeroPhobia as can be seen by the majority of them advancing through all VR exposure levels. Overall, fear levels decreased when participants progressed through the levels. Furthermore, our results suggest that it might be more beneficial to play one level for a longer period of time instead of practicing many VR levels and the existence of a ’sweet spot’, an optimum level of exposure at which increasing practice time does not result in increased benefit. In this study we found that participants derive most benefit when the practice time in the VR environment is 25.5 min irrespective of the amount of VR sessions. Finally, the importance of feeling present in the VR environment is stressed out as a higher reported presence was associated with better outcomes. Further research is needed to see if the gains from VR translate into long-term sustained results, both in virtual and in real-life situations.

## Figures and Tables

**Figure 1 jcm-09-01614-f001:**
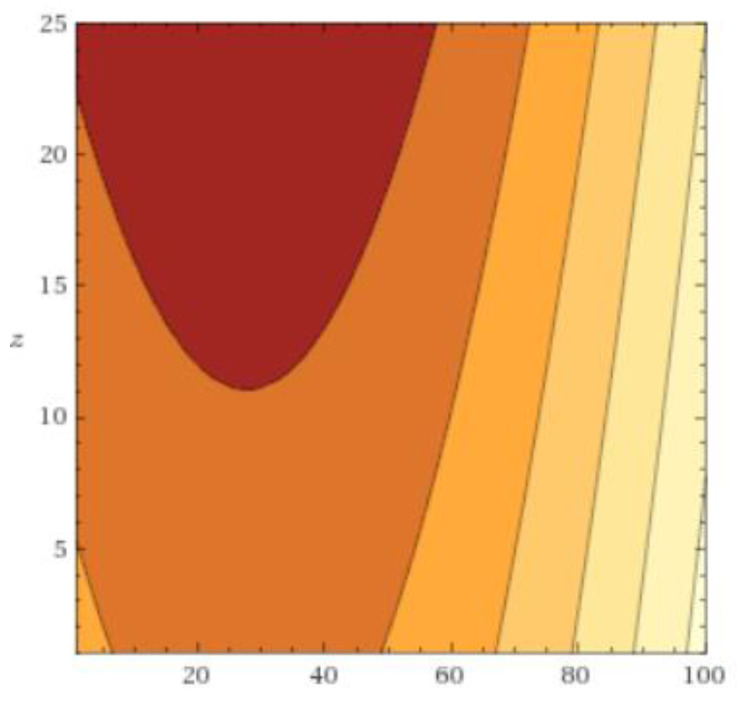
Practice Time (*x*-axis) vs. Number of Sessions (*z*-axis). Figure 1 visualizes the estimated function $AQ\_Total\_Post=-1.02x+0.02x^{2}-1.17z-12.85$ and plots a contour plot for x=1 to 100 and z=1 to 25. It thus represents a contour plot in which the outcome differences for the observed combinations of practice time (*x*-axis) and number of sessions (*z*-axis). The darker the color, the larger the outcome difference.

**Table 1 jcm-09-01614-t001:** Overall descriptive Statistics of Usage Data for Users who Experienced at least one VR session.

Variable	N	Mean	SD	Min	Max
Total					
Duration (total VR time in minutes)	66	24.384	16.127	0.600	71.050
Anxiety rating after playing level 1 ^1^	66	3.545	2.288	1.000	10.000
Anxiety rating of highest attained level ^2^	66	3.227	1.830	1.000	9.000
Anxiety rating after playing level first time ^3^	66	4.576	2.308	1.000	10.000
Anxiety rating after playing level last time ^4^	66	3.227	1.830	1.000	9.000
Number of VR sessions	66	9.167	5.365	2.000	25.000
Women					
Duration (total VR time in minutes) ^1^	47	23.516	15.744	0.600	58.967
Anxiety rating after playing level 1 ^2^	47	3.894	2.343	1.000	10.000
Anxiety rating of highest attained level ^3^	47	3.149	1.818	1.000	9.000
Anxiety rating after playing level first time ^4^	47	4.596	2.242	1.000	10.000
Anxiety rating after playing level last time ^5^	47	3.149	1.818	1.000	9.000
Number of VR sessions	47	9.085	5.344	2.000	24.000
Men					
Duration (total VR time in minutes)	19	26.532	17.289	3.517	71.050
Anxiety rating after playing level 1 ^2^	19	2.684	1.945	1.000	7.000
Anxiety rating of highest attained level ^3^	19	3.421	1.895	1.000	9.000
Anxiety rating after playing level first time ^4^	19	4.526	2.525	2.000	10.000
Anxiety rating after playing level last time ^5^	19	3.421	1.895	1.000	9.000
Number of VR sessions	19	9.368	5.560	3.000	25.000

^1^ Grubbs test was conducted to test for outliers and confirmed the apparent absence of outliers. Please see Appendix A for details on the variation in practicing duration. ^2^ anxiety rating after playing level 1 (the easiest level: changing a light bulb on a small ladder) for the first time. ^3^ anxiety rating after playing the highest level (for that person) for the last time, which is level 5 for most participants (range: 1–10). ^4^ anxiety rating after playing a level for the first time ^5^ anxiety rating after playing a level for the last time (range: 1–10).

**Table 2 jcm-09-01614-t002:** Level-specific Statistics of Usage Data for Users who Experienced at least one VR session.

Variable	N	Mean	SD	Min	Max
**LEVEL 1**					
Time spent in level (minutes) ^1^	65	2.289	2.109	0.317	11.483
Initial anxiety ^2^	65	3.554	2.305	1	10
Final anxiety ^3^	65	2.092	0.861	1	5
Number of sessions in level ^4^	65	2.046	1.556	1	8
**LEVEL2**					
Time spent in level (minutes)	62	5.322	5.334	0.283	36.350
Initial anxiety	62	4.129	2.229	1	10
Final anxiety	62	2.645	1.527	1	10
Number of sessions in level	62	2.177	1.337	1	6
**LEVEL 3**					
Time spent in level (minutes)	57	7.768	5.222	0.967	23.367
Initial anxiety	57	5.070	2.638	1	10
Final anxiety	57	2.877	1.489	1	9
Number of sessions in level	57	2.281	1.278	1	7
**LEVEL 4**					
Time spent in level (minutes)	50	6.742	5.639	1.100	29.517
Initial anxiety	50	4.700	2.517	1	10
Final anxiety	50	2.940	1.096	1	7
Number of sessions in level	50	2.260	1.724	1	11
**LEVEL 5**					
Time spent in level (minutes)	47	7.463	4.678	1.5	19.883
Initial anxiety	47	4.213	2.010	1	9
Final anxiety	47	2.872	1.035	1	7
Number of sessions in level	47	2.000	1.351	1	8
**TOTAL**					
Time spent in level (minutes)	281	5.727	5.114	0.283	36.350
Initial anxiety	281	4.302	2.397	1	10
Final anxiety	281	2.655	1.270	1	10
Number of sessions in level	281	2.153	1.450	1	11

^1^ Total time spent in this level, in minutes. ^2^ Self-reported anxiety after playing this level for the first time (range: 1–10).^3^ Self-reported anxiety after playing this level for the last time (range: 1–10).^4^ Number of sessions spent in this level.

**Table 3 jcm-09-01614-t003:** Mean decrease in self-reported fear after completing each session.

Variable	N	Mean	SD	Min	Max
Level 1	4	−3.500	3.873	−9	0
Level 1–2	6	−0.833	2.562	−6	1
Level 1–3	6	−0.167	2.562	−4	4
Level 1–4	11	−1.545	2.911	−6	4
Level 1–5	39	−1.333	1.644	−6	0
**TOTAL**	66	−1.345	2.243	−9	4

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
