# Peer review of "Analysis of Usage Data from a Self-Guided App-Based Virtual Reality Cognitive Behavior Therapy for Acrophobia: A Randomized Controlled Trial"

_jcm, 2020, doi:10.3390/jcm9061614_

Round 1

Reviewer 1 Report

The manuscript was improved to reflect the points I pointed out.

It is necessary to check the references of 11 and 12 in the reference list.

I have no other additional points.

Author Response

Dear Ms. Pei,

Our sincere thanks for your careful consideration of our manuscript Analysis of Usage Data from a Self-guided App-Based Virtual Reality Cognitive Behavior Therapy for Acrophobia: A Randomized Controlled Trial [ID] JCM 785674.  We would also like to thank the reviewers for their helpful comments and think that addressing them has significantly strengthened our paper. Below we have provided our itimized responses to the reviewers` comments.

We hope the revised version of the manuscript addresses all concerns raised. We would also be more than happy to make additional modifications, if desired. Thanks again for your consideration of our work.

Sincerely,

On behalf of the research team,

Tara Donker, PhD

Reviewer 1

Comment 1:

The manuscript was improved to reflect the points I pointed out. It is necessary to check the references of 11 and 12 in the reference list. I have no other additional points.

Reply: we thank the reviewer for pointing out the reference check. We have updated the references accordingly.

Reviewer 2 Report

The submitted paper is a companion to a previous paper, published in JAMA, and extensively referred to (we will come back to that). However, the idea of presenting, besides major outcomes, detailed usage data is quite interesting.

The original study is significant, showing that self-guided VR therapy is feasible and effective at home, with (relatively) low-cost equipement (if you assume that everyone can afford a powerful smartphone, which is debatable). 

Besides the undeniable interest of this research for readers, there are a number of points that need to be addressed before this paper be suitable for publication. At present, it looks more like a working draft.

First, there are many places in the paper where the reader is sent to the original (JAMA) paper (which does not give the reader enough information to easily understand results. 

For example, The Acrophobia Questionnaire is barely described, and the reader does not know (later) where data come from and how they were analysed. 

More disturbing is the fact that IPQ no longer appears in abstract, whereas it is evoked in introduction and results are reported later. It must as well be shortly described.

In other words, serious editing of this paper is required. 

In more details, throughout the paper

The introduction is not focused on the interest of usage data analysis. More precisely, it is much less informative than the JAMA intro of VR therapy, and should start more directly to the interest of usage data analysis in self-guided VR.

In materiels and methods, description of the "virtual therapist" is missing (Is it a voice, an avatar?). Even if the soft is commercial, a few screenshots might help. On this point, referring to supplemental material of the first paper is not fair (line 167).

Concerning gaze, it is written that it is used for locomotion control. In the same time, participants were asked to look for items. How did that work?

Also, 'the risk for cybersickness sas minimized by optimizing the framerate". More info please.

Concerning security, how would an incident be dealt with?

Concerning statistics, it not clear why the authors changed their model (line 202). 

In results, besides the fact that the sickness questionnaire (SSQ) is not described (shortly), the way its outcome (less than 3 symptoms) is not canonical. Justify please.

Concerning VR usage data, the variability is huge (data are what they are!) and no wonder statistical analysis is not convincing.

I am still surprised that, in table 1, duration goes from .6 to 71 minutes. A description of the distribution might be helpful here. Outliers? 

How did you process data to get figure 1? It is incorrect to write that this figure has 3 axes. 

A more formal comment: Why do hypotheses appear here? We would expect them in the introduction. 

Even if modeling has changed (see above, why?), hypothesis 1 was validated in the first paper. 

Last point, in conclusion, where does the 25.5 minutes come from?

Hope these comments help.

One last comment: besides the fact that the application seems to "work", the great inter-subject variability in usage is remarkable. The authors mention a small sample size as resulting in poor statistical power. One alternative view might be that, depending on the personal history, VR stimulation will work at one moment of exposure, without a tool to identify the triggering event, conditioning success of fear reduction. 

Author Response Letter

Manuscript: Analysis of Usage Data from a Self-guided App-Based Virtual Reality Cognitive Behavior Therapy for Acrophobia: A Randomized Controlled Trial

Reference number: JCM 785674

Dear Ms. Pei,

Our sincere thanks for your careful consideration of our manuscript Analysis of Usage Data from a Self-guided App-Based Virtual Reality Cognitive Behavior Therapy for Acrophobia: A Randomized Controlled Trial [ID] JCM 785674.  We would also like to thank the reviewers for their helpful comments and think that addressing them has significantly strengthened our paper. Below we have provided our itimized responses to the reviewers` comments.

The revised manuscript text now comprises 5715 (including tables and headings). The amount of Figures and tables did not change compared to the previous version. We have included one extra supplementary file (supplementary file 2). In addition to changes made based on the suggestions of the reviewers, we have deleted or rephrased sentences throughout the manuscript.

We hope the revised version of the manuscript addresses all concerns raised. We would also be more than happy to make additional modifications, if desired. Thanks again for your consideration of our work.

Sincerely,

On behalf of the research team,

Tara Donker, PhD

Reviewer 1

Comment 1:

The manuscript was improved to reflect the points I pointed out. It is necessary to check the references of 11 and 12 in the reference list. I have no other additional points.

Reply: we thank the reviewer for pointing out the reference check. We have updated the references accordingly.

Reviewer 2

Comment 1:

First, there are many places in the paper where the reader is sent to the original (JAMA) paper (which does not give the reader enough information to easily understand results. For example, The Acrophobia Questionnaire is barely described, and the reader does not know (later) where data come from and how they were analysed. More disturbing is the fact that IPQ no longer appears in abstract, whereas it is evoked in introduction and results are reported later. It must as well be shortly described. In other words, serious editing of this paper is required. 

Reply: We have added relevant information that enables readers to understand this paper without having to consult the JAMA paper. To this end we have added  a description of the Acrophobia questionnaire (page 5, line 184): “The primary outcome was the 20-item Acrophobia Questionnaire (AQ) [27]. The AQ is a widely-used validated instrument [27]. The anxiety subscale is measured using a 7-point Likert scale (0=not anxious to 6 = extremely anxious). The total score range is 0-120. The avoidance subscale uses a 3-point Likert scale (“I would not avoid it” to “I would not do it under any circumstances”). 

The usage data is described in the Method section on pages 4 and 6 of the manuscript, (e.g. study design and usage data retrieval). We have elaborated a bit more on usage data retrieval: “participants were recruited from the Dutch general population through websites, magazines and local media”, “All materials were completed online without researcher intervention”  and “Usage data was retrieved from the ZeroPhobia app to a server which then stored the data on a database, on a secure (SSL) website( …) Examples of database reads are time spent in a VR level, time stamps (start and end time of a participant in the VR environment) that enabled us to determine the exact duration of practice time for each VR level, and experienced anxiety levels after each VR level played”. The Level-specific Statistics of Usage Data for Users who Experienced at least one VR session are shown in Table 2.

We apologize for the fact that we mistakenly removed the IPQ from the abstract. This was unintended and we have included it in the Abstract again. We thank the reviewer for bringing this to our attention.

Comment 2:

The introduction is not focused on the interest of usage data analysis. More precisely, it is much less informative than the JAMA intro of VR therapy, and should start more directly to the interest of usage data analysis in self-guided VR.

Reply: Although we see the reviewer’s point here we would like to emphasize that the topic of this special issue concerns innovative technology-based interventions for common mental disorders. The rationale behind starting with a very brief background on VR therapy is, in line with the purpose of the special issue, to inform readers about the importance of innovative technology-based interventions, such as VR therapy. In turn, stressing the value and importance of VR therapy in the field of psychological treatment helps clarify to readers why it is important to examine its usage data.

Comment 3:

In materiels and methods, description of the "virtual therapist" is missing (Is it a voice, an avatar?). Even if the soft is commercial, a few screenshots might help. On this point, referring to supplemental material of the first paper is not fair (line 167).

Reply: we have added the underlined words to the following sentence on p. 4 line 135: “ZeroPhobia-Acrophobia consists of six animated and engaging modules using 2D animations which provide background information and explain key concepts (e.g. the fear curve). The animations are accompanied by an explanatory voice-over and an animated virtual therapist, modelled after the first author”

The screenshots were included in the current paper: “..see for ZeroPhobia screenshots supplementary file 1 (page 5, line 180)”. To be sure, we did not refer to supplementary material of the first paper.

Comment  4:

Concerning gaze, it is written that it is used for locomotion control. In the same time, participants were asked to look for items. How did that work?

Reply: We have added the following text to clarify this in the manuscript (page 4 line 143): “Gaze control is a method for the hands-free selection of objects and the activation of functions within a virtual environment. By looking at an object or button in the center of the field of view for a specified period of time, e.g., two seconds, the object or button is selected or activated. In ZeroPhobia, large arrows served as interactive buttons used for navigating the virtual environment. By gazing at an arrow, a user moved to the location of the arrow. Similarly, items that needed to be collected,  as part of the assignments in the various levels, could be selected by looking at them.”  

Comment  5:

Also, 'the risk for cybersickness was minimized by optimizing the framerate". More info please.

Reply: Because any VR setup that generates frame rates below 90 frames per seconds is likely to induce disorientation, nausea and other negative user effects, framerate was increased to an optimal level to minimize the risk of cyber sickness. This has been added to line 160, page 5.

Comment  6:

Concerning security, how would an incident be dealt with?

Reply: We assume the reviewer is referring to an incident of harm with a user instead of an incident with security of data from the ZeroPhobia app (but please correct us if we are wrong). In the unlikely case of an incident, the procedure for incidents as described in the medical ethical report would have been followed. According to this procedure,  users are instructed to inform the research team in the case of an incident. In turn, again following procedures, the medical ethical committee would be informed. In case of physical or psychological damage, VU University would provide coverage for adequate medical care.    

Comment  7:

Concerning statistics, it not clear why the authors changed their model (line 202). 

Reply: The model referred to in line 202 is equivalent to the model estimated in the original JAMA paper, but the main effect is now interacted with background characteristics. To be more precise, the maximum likelihood estimation model generates equivalent estimation results to the original OLS model, but is more convenient to estimate and to infer interaction terms.  

Comment  8:

In results, besides the fact that the sickness questionnaire (SSQ) is not described (shortly), the way its outcome (less than 3 symptoms) is not canonical. Justify please.

Reply: We have deleted the results from the SSQ because this was not a focus of the current paper.

Comment  9:

Concerning VR usage data, the variability is huge (data are what they are!) and no wonder statistical analysis is not convincing.

Reply: We have now given a detailed description of the data (see reply 1) and agree that the number of VR levels played and minutes spent in the virtual environment differs substantially between participants. The histogram below shows this variation in practicing duration but does not reveal outliers (we come back to this in our reply to comment 10). Table 2 shows descriptive information on the levels played, but the table does not indicate that the observed variability or variance represent outliers or strange values.

Practice Duration

Comment  10:

I am still surprised that, in table 1, duration goes from .6 to 71 minutes. A description of the distribution might be helpful here. Outliers? 

Reply: The table reflects practice duration in minutes. The plot that shows Duration in minutes for Users who Experienced at least one VR session  shows that the distribution is skewed to the right, but is nevertheless smooth and continuous (implying there is no reason to suspect that a certain observation is an outlier). A formal test for detecting outliers relies on the normality assumption. Notwithstanding this caveat, Grubbs’ test confirms the apparent absence of outliers.

Grubbs F. (1969), Procedures for Detecting Outlying Observations in Samples, Technometrics, 11(1), 1-21.

We have noted the formal Grubbs test for outliers in the footnote of Table 1 and included the above histogram as a supplementary file.

Comment  11:

How did you process data to get figure 1? It is incorrect to write that this figure has 3 axes. 

Reply: Thank you for this comment. We fully agree that this figure does not have 3 axes. The Figure visualizes the estimated function  and plots a contour plot for  and It thus represents a contour plot in which the outcome differences for the observed combinations of practice time (x-axis) and number of sessions (z-axis). The darker the color, the larger the outcome difference. We have now relabeled Figure 1 as follows:

Figure 1. Practice Time  (x-axis) vs. Number of Sessions (z-axis).

Comment  12:

A more formal comment: Why do hypotheses appear here? We would expect them in the introduction. 

Reply: The hypotheses are provided in the introduction (see page 4). The hypotheses are repeated in the result section for reasons of clarity and structure.

Comment  13:

Even if modeling has changed (see above, why?), hypothesis 1 was validated in the first paper. 

Reply: In the current paper, hypothesis 1 concerns: “More VR activity is associated with lower post-test AQ scores”. We did not include this hypothesis in the first paper (as no usage data was analyzed in that paper) and instead tested the hypothesis that “the app would be associated with greater overall response at post-test compared with a wait-list control group”. Perhaps the reviewer wonders why we provided the reader with a short replication of the main results. We deemed this necessary because the background characteristics from the usage data interacted with the estimated main effects.  The estimation models in both papers are equivalent, but a maximum likelihood procedure is more convenient to estimate and to infer interaction terms (See also the reply to comment 7).

Comment 14:

Last point, in conclusion, where does the 25.5 minutes come from?

Reply: This was an outcome from hypothesis 1, Figure 1. We have now explicitly included this to the revised paper, page 9, line 298:  The optimum level of practice time in the VR environment was found to be 25.5 minutes.

Comment  15:

One last comment: besides the fact that the application seems to "work", the great inter-subject variability in usage is remarkable. The authors mention a small sample size as resulting in poor statistical power. One alternative view might be that, depending on the personal history, VR stimulation will work at one moment of exposure, without a tool to identify the triggering event, conditioning success of fear reduction. 

Reply: We thank the reviewer for pointing this out and have included this in the discussion section.

Round 2

Reviewer 2 Report

Thanks to the authors for addressing my concerns. I will approve the paper for publication in its new form.

This manuscript is a resubmission of an earlier submission. The following is a list of the peer review reports and author responses from that submission.

Round 1

Reviewer 1 Report

Dear Authors, 

Your research on usage data from a self-guided app-based VR cognitive behavior therapy was very clear, thorough and organized. The research problem is well presented as well as the experimental design data analysis and contributions. This work is highly relevant as supported by the literature review and current trends in the field.

In the attached PDF you will find some comments and suggestions. My main concern is regarding the materials and methods, which you have cited. I believe that providing a brief description of the virtual reality environment can help readers better understand what was presented to the participants, which can cause a major positive impact on the conclusions.

Reviewer 2 Report

This study developed its own self-guided VR program to find out its effectiveness. It was aimed at adults with fear of heights, and evaluated the fear of heights by producing self-guided VR programs. Similarly, previous studies (Hong et al., 2017) already showed that mobile-based self-training VR program for feat of height were the first to be invented and investigated its effectiveness. Although it was very similar to this study, it did not appear to be covered in this paper at all so that it was needed to be mentioned in the text. Unlike Hong et al's study in which a subject attended a VR-Center and performed experiments in a controlled environment, this study was meaningful in that it used the data which was collected by participants being at home autonomously. However, it is quite hard to consider it proved the effectiveness of self-guided VR program for acrophobia reasonably for the following reasons, including the key drawback of insufficient statistical evidence.

Lack of systematic indicators related to fear of heights

First, since this study was not conducted by a subject in a controlled environment, there may be a question of how to ensure data quality before evaluating the effectiveness of a self-guided program. In other words, data collected through autonomous participation requires more detailed variables. To address this problem, there should have been more systematic indicators specific to fear of heights than general indicators (i.e. number of VR completed session, practice time in VR) or simple subjective fear ratings that already have been addressed in previous VR studies. For example, Hong et al. (2017) analyzed these data by embedding physiological indicators such as Heart Rate (HR) and behavioral indicators such as gaze down percentage in the APP. These data can be more objective because they are not consciously recognized by users. In fact, in addition to the research aspect, users can actually see the performance feedback report using this objective result directly in the app, which has a positive effect on self-training. Therefore, advanced variables are needed to function not only as a guarantee of data quality, but also as a training app specialized for fear of heights. This paper mentioned that 'interactive VR environment was used and users navigated the environment using gaze control (line 122-123)', so it could be a good way to analyze data related to the gaze control.

Lack of references related to hypotheses

Five hypotheses were presented in the text, but no relevant prior studies or evidence were given. The hypothesis should be based on existing research and be able to support it. In particular, the hypothesis of lines 90-92 (4) “higher acrophobia symptoms at pre-test correlate positively with VR activity (number of completed VR sessions, practIce time in VR), and negatively with a reduction in acrophobia symptoms at post-test” does not make sense when referring to existing studies and appears to be an insufficient evidence. This is because people with higher fear of height are more likely to exhibit avoidance behavior during VR experience, and actually show negative correlation with Gaze-down percentage in previous papers (Hong et al., 2017). In addition, it is contrary to the hypothesis of this paper because Hong (2017) actually confirmed that higher fear of height was positively correlated with a reduction in acrophobia symptoms at post-VR training.

Problem with Sample

1) Absence of diagnostics for Acrophobia

Having a fear of height and acrophobia are considered to be a slightly different concept. Although many people may have a Fear of height, strict clinical diagnosis is required to mark them as acrophobia. Therefore, the AQ score 45.45 used in this paper is a difficult criterion to consider as acrophobia, and even there is no reference to the reason for adopting 45.45 for the criterion. Similarly, in Hong et al (2017), the standard 45.45 was taken from previous papers to divide relative high- and low-groups of fear of height for general volunteers. However, it is unreasonable to conclude that all those who score above 45.45 as acrophobia as in this paper. In line with this, when looking at Table 1, because there are people with the lowest anxiety rating as 1 (range 1 to 10) for the every item, those people would be difficult to consider them as acrophobia (as 1 is close to the level without fear of heights).

2) The problem of gender ratio

According to Line 177, 66 subjects were women, accounting for 68.75% of all participants. Therefore, it is necessary to verify whether there is no significant difference in the result value between men and women, because some previous studies report differences between men and women as for experiencing fear of height.

3) Problems with Sample Number

To sum up Line 179-186, 96 participants participated, but 21 failed to download the app and 1 failed to experiment due to health problems, so only 74 participants succeeded in running the app. Of the 74, only 66 completed at least one VR session (finally, only 47 completed all VR levels). Therefore, to prevent readers from confusion, it is advisable to exclude people who did not technically or personally run the app from the research sample counting. In conclusion, the number of people who participated in VR training should be marked as 74, and it seems necessary to modify it on abstract.

High drop-out rate and absence of safety measurements

74 people started the VR experience, but only 66 experienced at least one VR session. In other words, only 66 people were significantly trained in VR. Therefore, it is necessary to review the reasons for the drop-out of eight people who failed to complete at least one VR session. In addition, only 47 of the 74 patients finally completed all levels (27 of 74 drop-outs are not small). To sum up, the text didn't pay any attention to 8 (of 74) who didn't finish at least one session, and it is also unappropriated to assume another 9 drop-out (of 66) who started at least one session but didn't finish all levels, would be totally due to fear of heights. This high drop-out rate may be closely related to the discomfort associated with simulator sickness. It is very likely that simulator sickness will be easily induced due to the perspective of the virtual space with the height of fear. In addition, this study uses 3D animation that requires high-loading, so the screen is difficult to be compatible with the user's movement in real time, which may cause visual latency inducing simulator sickness. In fact, people with heights are known to experience more simulator sickness. Therefore, it is necessary to measure the scale such as Simulator Sickness Questionnaire (SSQ) as in Hong et al (2017). In the Simulator Sickness Questionnaire (SSQ), there are three subscales, nausea, oculomotor, and disorientation, which can be used to examine in detail which factors caused drop-out and related to treatment effectiveness inhibition. (Actually, a paper by Hong et al (2017) shows that oculomotor, or eye strain, inhibited treatment efficacy).

Since self-guided VR programs should be performed by individuals, assessment of the effectiveness of training programs should always be accompanied with these safety assessments, especially when acrophobia is a target user. Therefore, there is an obligation to prove that the high dropout rate in this study does not result from safety problems. Also, there are no safety measurement, so line 334 “results shows that the VR environment offers safe and engaging way to self-guided exposure” should be revised. In addition, some subjects may have intentionally chosen a fear rating of 0 to 4 to move on to the next environment due to the curiosity about the next level of the virtual environment, or they might have lost interest in the VR program and made a drop-out midway. Therefore, it may be related to the first discussion point that another objective indicators are needed to guarantee data quality.

Lack of Statistically Significant Results

Most importantly, this paper argues about the effectiveness of the program, but does not provide sufficient statistically significant results. Hypotheses should be demonstrated and discussed on the basis of significantly different outcomes, not just descriptive differences. With regard to overall usage data and level specific data, only descriptive information on mean, SD, minimum and maximum observations is presented and it is not known whether there is a statistically significant difference. Therefore, it is difficult to verify the effectiveness before and after training. The only statistical test stated that there was a statistical difference between anxiety rating for the first time and anxiety rating for the last time, averaged across all level environments (line 202), even this is not shown in the presented table. The only other statistical result presented is “the pre-test AQ scores were significantly associasted with the subject fear rating after the first session (r = 0.134, p <.05) and a reduction in AQ score at post-test (r = -0.312, p <.05) ”(line 261-265). However, this finding is exactly the same as the finding presented in Hong et al (2017), so there is nothing new.

Incomplete Results and Interpretation with regard to Figure 1

1) Incomplete results

The title description in Figure 1 states ‘Session length’ (x-axis), which is described as ‘practice time’ (line 229). This value is expected to be the same as ‘Duration (Total VR time in minutes)’ presented in Table 1. Therefore, the names should be unified so as not to be confused. Strangely enough, in Table 1, the duration ranges from Min 0.6 to Max 71.05. However, Figure 1 shows the range of X values from 0 to 100. Please check again on this part.

2) Incomplete Interpretations

Regarding Figure 1, the text argues that: “It is more beneficial to play one level for a longer period instead of practicing the same level repeatedly” (line 35-36), “These results suggest the existence of a 'sweet spot', an optimum level of exposure at which increasing practice time does not result in increased benefit ”(line 231-232 & line 327-332). However, this interpretation is misleading and should be interpreted more specifically as follows: For example: 1) 'the training tends to be effective when the total practice time is less than 60 min', 2) 'the training tends to be effective regardless of the number of sessions when the total practice time is close to 30 minutes, and 3) 'If the total practice time is very short or close to 60 minutes, the training tends to be effective for as many repeated exposures as possible.'

In particular, “it is more beneficial to play one level for a longer period instead of practicing the same level repeatedly” (line 35-36) is a hasty generalization and underestimates the repeated exposure effect. In general, Cognitive Behavior Therapy (CBT) affirms the therapeutic effect of repeated exposure, and several studies have demonstrated anxiety reduction effects through repeated exposure to the same VR environment. In addition, a recent study on a similar self-training program (Hong et al, 2017) found that there was a time effect on subjective fear rating and heart rate (HR) when first exposure and second exposure for the same environment were compared. It was also demonstrated that the effects of this repeated exposure was significantly greater in the High fear group than in the Low fear group. Thus, there seems to be a lack of accurate interpretation and sufficient discussion based on the data. 

Insufficient description of virtual environment setting

The virtual environment scenario is a key element of the VR training program, but the explanation is insufficient. It was suggested that the total VR experience time took 24.4 minutes (min 0.6, max 71.05) on average (table 1), which indicates a large variation among subjects. In this regard, it is necessary to accurately state the length of a video presented on VR (or whether a certain length of video is repeatedly played). At the same time, there is a lack of explanation of the sound setting. Sound is an important factor in making the virtual environment feel real. If you provide only a visual virtual environment without wearing an ear-phone, this will distract from the auditory and visionary experiences and seriously impair reality. Although this study has the above limitations, this is a noteworthy study in that 1) it assessed the outcome of presence questionnaire that has been absent at previous papers in regard with fear of height, 2) it used low-cost VR goggles (cardboard goggles) enabling generals to access it easily forward, and 3) it addressed data-driven ‘sweet spot’ related to training effect bringing a new discussion point that has not been covered in this field.